# Accumulation of Proline in Plants under Contaminated Soils—Are We on the Same Page?

**DOI:** 10.3390/antiox12030666

**Published:** 2023-03-08

**Authors:** Sofia Spormann, Pedro Nadais, Filipa Sousa, Mafalda Pinto, Maria Martins, Bruno Sousa, Fernanda Fidalgo, Cristiano Soares

**Affiliations:** 1GreenUPorto-Sustainable Agrifood Production Research Centre & Inov4Agro, Faculty of Sciences, University of Porto, Rua do Campo Alegre s/n, 4169-007 Porto, Portugal; 2Biology Department, Faculty of Sciences, University of Porto, Rua do Campo Alegre s/n, 4169-007 Porto, Portugal

**Keywords:** oxidative stress, redox homeostasis, antioxidants, soil erosion, abiotic stress, salinity, metals, xenobiotics

## Abstract

Agricultural soil degradation is occurring at unprecedented rates, not only as an indirect effect of climate change (CC) but also due to intensified agricultural practices which affect soil properties and biodiversity. Therefore, understanding the impacts of CC and soil degradation on plant physiology is crucial for the sustainable development of mitigation strategies to prevent crop productivity losses. The amino acid proline has long been recognized for playing distinct roles in plant cells undergoing osmotic stress. Due to its osmoprotectant and redox-buffering ability, a positive correlation between proline accumulation and plants’ tolerance to abiotic stress has been pointed out in numerous reviews. Indeed, proline quantification is used systematically by plant physiologists as an indicator of the degree of tolerance and a measurement of the antioxidant potential in plants under stressful conditions. Moreover, the exogenous application of proline has been shown to increase resilience to several stress factors, including those related to soil degradation such as salinity and exposure to metals and xenobiotics. However, recent data from several studies often refer to proline accumulation as a signal of stress sensitivity with no clear correlation with improved antioxidant activity or higher stress tolerance, including when proline is used exogenously as a stress reliever. Nevertheless, endogenous proline levels are strongly modified by these stresses, proving its involvement in plant responses. Hence, one main question arises—is proline augmentation always a sign of improved stress resilience? From this perspective, the present review aims to provide a more comprehensive understanding of the implications of proline accumulation in plants under abiotic stress induced by soil degradation factors, reinforcing the idea that proline quantification should not be employed as a sole indicator of stress sensitivity or resilience but rather complemented with further biochemical and physiological endpoints.

## 1. Introduction

Climate change (CC), environmental pollution, and the loss of agricultural productivity are major issues of our time, particularly because they are interconnected and aggravate one another and even more so for being the most threatening to regions of the planet that are already vulnerable [1,2,3]. The degradation of agricultural soils follows at unprecedented rates, not only due to the indirect effects of CC but also due to the use of heavy machinery, excessive and negligent irrigation, and the application of agrochemicals, which altogether lead to soil salinization and contamination and a loss of soil functions and biodiversity [4,5]. Thus, in a snowball phenomenon, large-scale agricultural production on increasingly infertile soils tends to accelerate soil erosion, indirectly contributing to the worsening of CC [5,6,7]. In this sense, it is imperative to reinvent modern agriculture, reduce its ecological footprint, and make it more resilient to these inevitable adversities [4]. Understanding the effects of CC and soil degradation on plant physiology is of the utmost importance in the research on sustainable mitigation approaches to avoiding crop productivity losses. Abiotic stress mitigators have gained attention from plant researchers, including those that aim for a tolerance to the stresses induced by soil degradation, such as salinity, metal toxicity, and exposure to xenobiotics [8]. Several compounds have been proposed as stress mitigators, including biostimulants, plant-growth-promoting microorganisms, phytohormones, trace elements, N-containing compounds, and the focus of this work: the amino acid proline [9,10,11]. In fact, the exogenous application of proline as a stress mitigator is evidenced in numerous reviews, having been used to boost plant tolerance to several abiotic stresses [10,12,13,14,15,16,17,18,19,20].

Taking advantage of the already extensive literature regarding the participation of proline in plant responses to environmental stress, the present work aims to further disclose the roles of proline in the specific cases of plant abiotic stress related to soil degradation. Providing a broader perspective, the proline metabolism responses of plants under exposure to excessive salt, metals, and xenobiotics are described below. Examples of studies in which proline accumulation was used for the mitigation of stress, either by exogenous application or by employing genetic engineering tools, are also presented. The suggested roles and implications of proline accumulation in stress signaling and tolerance are critically discussed, highlighting how this amino acid appears to have such divergent functions in cells under stress depending on the type of stress and the plant species, age, and tissue, as well as the source of proline itself.

## 2. Proline: Functions and Metabolism

Proline is a proteinogenic amino acid with a secondary amine in the α-amino group and an unusual cyclic structure which confers conformational rigidity to the secondary structure of the proteins [18,21]. In addition to taking part in the composition of proteins, free proline has also been recognized for playing distinct roles in plant cells, especially under conditions of abiotic stress. In addition to occurring as a product of proteolysis, proline can be biosynthesized *de novo* in plant cells by the glutamic acid (Glu) pathway or, alternatively, by the ornithine (Orn) pathway (Figure 1). L-proline is mainly biosynthesized from Glu in the cytoplasm and/or chloroplast through the activity of two enzymes: Δ1-pyrroline-5-carboxylate-synthetase (P5CS, EC 2.7.2.11; EC 1.2.1.41) and Δ1-pyrroline-5-carboxylate-reductase (P5CR; EC 1.5.1.2). The first is responsible for the conversion of Glu into glutamic-5-semialdehyde (GSA), which then undergoes a spontaneous cyclization to form pyrroline-5-carboxylate (P5C). P5CS is encoded by two genes whose expression is differently regulated, resulting in two isoforms which are usually present in non-redundant activity, have different subcellular locations, and are reportedly sub-functionalized in their stress-responsive (*P5CS1*) and housekeeping (*P5CS2*) forms [10,18,20,22,23]. On the second enzymatic step of this biosynthetic pathway, P5CR reduces P5C into proline, preferentially using NADPH as the electron donor. In the alternative pathway, the first step encompasses the reversible transamination of Orn into GSA, which is catalyzed by the mitochondrial enzyme ornithine δ-aminotransferase (OAT; EC 2.6.1.13) (Figure 1). Similar to the Glu pathway, GSA is then spontaneously cyclized into P5C, which is converted into proline by P5CR [18,24]. The catabolism of proline takes place in the mitochondria through the activities of the enzyme proline dehydrogenase (ProDH; EC 1.5.5.2), which converts proline back to P5C. This common intermediate, P5C, is oxidized by P5C dehydrogenase (P5CDH; EC 1.2.1.88) to produce Glu. The enzyme ProDH also comes in two isoforms, which are expressed under different circumstances [25]. Although the exact mechanisms of intracellular transport are not yet well described, the compartmentalized metabolism of proline involves the active transport of proline, P5C, GSA, and Glu between the cytosol, chloroplasts, and mitochondria [26] (Figure 1). When P5C oxidation in the mitochondria is impaired, P5C can be exported to the cytosol and be reduced to proline by cytosolic P5CR, forming the recently proposed proline–P5C cycle [10,12,18,24,27,28]. A review by Ben Rejeb et al. [28] mentioned that when cells accumulate excess proline, the proline–P5C cycle is more active and more electrons are delivered to the mitochondrial electron transport chain, leading to a greater production of reactive oxygen species (ROS). For this reason, P5C has been recognized as toxic due to the induction of oxidative stress, and the process of proline degradation in plants has been associated with the generation of hypersensitive responses (HRs) and programmed cell death (PCD) [28]. The levels of free proline inside the cells are regulated by the balance between biosynthesis and degradation. Under normal conditions, the proline degradation pathway is positively regulated by its accumulation in a feedback mechanism. In fact, plants treated with exogenous proline usually show an induction of proline oxidation via the activity of ProDH and P5CDH in the mitochondria [29].

Proline has been reported to play important roles in the developmental processes associated with plant reproduction. Under normal conditions, the highest levels of proline in a plant are found in the flowers, especially in pollen grains, and in the seeds. The lowest levels are found in the roots [30]. Proline seems to be required as an osmoprotectant during pollen development and embryogenesis, which are both desiccation-dependent processes [22]. Moreover, proline has also been found to accumulate in actively dividing meristematic tissues, such as the root tip, shoot apex, lateral buds, inflorescence, and the germinating seed, acting as a source of energy to support such metabolically demanding programs. Indeed, during proline catabolism in the mitochondria, the oxidation of one molecule of proline yields 30 ATP [22,31]. Regarding the role of proline in growth, in addition to providing the energy required for cell division, proline is also involved in cell elongation as the extracellular proteins hydroxyproline-rich O-glycoproteins (HRGPs) are important components of the plant cell wall structure and contain variations of proline–proline repeats [32]. The light-dependent reciprocal gene activation of P5CS has also been reported to cause a light-dependent accumulation of proline, showing a strong relationship between photosynthesis and proline metabolism [19]. In the absence of stress, the biosynthesis of housekeeping proline seems to take place in the cytosol because of the activity of *P5CS2* [19]. 

Most studies on this subject report the accumulation of proline in plant cells during events of abiotic stress, especially under conditions of osmotic stress (Figure 2). Its high solubility in water and zwitterionic state allow proline to accumulate in high amounts inside cells [18], a property that proline shares with other compounds commonly known as compatible solutes, which are also built up by an array of organisms to adjust cellular osmolarity under conditions of osmotic stress [18]. Thus, proline is mostly recognized for its role as a potent osmoregulator, namely, to protect cells by increasing water uptake under conditions of low water availability and salinity stress. In addition to its role as a compatible solute, proline is also widely recognized for improving the stability of proteins and the integrity of membranes, facilitating the activation of enzymes and serving as a redox buffer, ROS scavenger, and metal chelator in the cytoplasm. Interestingly, aside from these functions, proline has been shown to act as a signaling molecule [18]. This is generally represented in Figure 2 and will be described in more detail throughout the review.

## 3. How Is Proline Homeostasis Affected by Stress?

The induction of proline accumulation under stress usually occurs due to the increased *de novo* synthesis and does not seem to rely as much on proteolysis events [29]. The reduction in turgor is probably the main trigger for proline accumulation in plants under osmotic-related stresses such as drought, heat stress, and salinity [29]. These conditions usually induce stomatal closure, limiting carbon uptake and subsequently decreasing NADPH consumption by the Calvin cycle, depleting the pool of the electron acceptor NADP^+^. This leads to singlet oxygen (^1^O_2_) production and the accumulation of ROS [28]. Under abiotic stress, there is an activation of the biosynthetic enzyme *P5CS1* but not of *P5CS2* [30]. There is also a stimulated import of *P5CS1* and P5CR enzymes into the chloroplasts, directing the buildup of proline to these organelles during stress [15,19,28,30,33]. Abiotic stress has also been shown to inhibit proline degradation pathways through the inhibition of the ProDH and P5CDH enzymes at expression and protein activity levels [20,30]. This overrides the feedback inhibition mechanism in order to maintain high levels of the newly synthesized proline. Moreover, plants under stress also show changes to the permeability of the mitochondrial membrane that prevent proline from reaching its site of oxidation. They also demonstrate an impaired incorporation of proline into proteins to maintain high levels of free proline in the cytoplasm and chloroplasts [29].

The spatial distribution of proline seems to play an important role in the functional diversification of this amino acid [34] (Figure 1). For instance, the compartmentalized biosynthesis of proline in chloroplasts is light-dependent and is stimulated during osmotic stress. It is associated with the protection of the photosynthetic apparatus [19] as it is a reductive pathway that requires NADPH to reduce Glu and P5C, restoring the pool of NADP+ and sustaining the electron flow between photosynthetic excitation centers. In this way, proline synthesis may lower the production of ROS in the chloroplasts to help maintain redox balance and avoid oxidative damage and photoinhibition. 

Verbruggen and Hermans [30], Kavi Kishor and Sreenivasulu [27], and others have highlighted that the stress-induced inhibition of proline oxidation ceases once the stress is relieved. The pools of stress-induced proline supply a reduced potential for the mitochondria through the oxidation of proline by ProDH and P5CDH, providing electrons for the respiratory chain [19]. Once stress is relieved, the degradation of the highly accumulated proline provides a significant supply of energy that could be used by cells to resume growth [19,27]. In addition, during the imposition of drought and/or salinity stress, some proline has been shown to be translocated from the shoots to the roots, where its subsequent degradation is thought to be used as an energy reservoir that could support root growth, allowing the roots to reach deeper in the soil profile in the search for areas of higher moisture [27]. This raises an interesting question on the involvement of proline in the perception of stress, suggesting that the catabolic enzymes ProDH and P5CDH are not as inhibited by stress in the root tissues as they are in the leaf tissues, where proline is so promptly synthesized.

## 4. The Involvement of Proline in Plants Facing Issues of Soil Degradation

### 4.1. Salinity Stress

Soil salinity problems have accompanied humanity throughout known history, with historical reports showing the downfall of many civilizations due to the inability of agriculturally based societies to deal with a loss of soil viability through salinization [35]. Due to natural and anthropogenic factors, either at an individual (e.g., the rise of seawater levels, excessive fertilization, or poor management practices) or synergic (e.g., reduced precipitation and excessive irrigation) level, over 1.5 Mha of arable land is becoming unsuitable for crop production per year, with up to half of all cultivable lands throughout the world being expected to become inviable for agriculture by 2050 [6,36]. 

Indeed, it is well known that exposure to a high salt content severely affects crop yield by decreasing the plants’ water uptake, disrupting the ionic balance, inducing oxidative stress, and limiting photosynthetic efficiency, ultimately leading to decreased productivity or even the death of the plants [37]. For these reasons, extensive research has been conducted on the physiological effects of salinity stress and uncovering possible tolerance traits.

#### 4.1.1. Role of Proline in Salt-Stressed Plants

Although salt stress can affect plants in different ways, one of the most common responses is an exacerbated accumulation of proline [38,39,40,41,42,43,44]. Plant researchers have been dedicated to understanding whether this ubiquitous increase in proline content could be explored as a sign of tolerance or as a sign of stress susceptibility. Despite this, and as further reviewed below, a consensus is yet to be reached regarding the role of proline accumulation under saline conditions. 

Although some authors show a preference for the role of the OAT pathway in mediating proline synthesis under salt stress [45], most works to date report a high upregulation of the P5CS pathways [46,47,48,49,50] that is sometimes coupled with a slight increase in OAT activity [51,52]. Curiously, Wang et al. [53] showed that when *Saussurea amara* (L.) DC. plants were exposed to NaCl concentrations ranging from 100 to 400 mM, P5CS takes the leading role in proline biosynthesis only up to 200 mM, with the increased proline accumulation under higher salt concentrations being ascribed mainly to OAT. Contrasting results were found by Bagdi et al. [54] in which an OAT pathway in a salt-tolerant variety of rice was induced in a generalized manner but P5CS was only induced for the highest NaCl concentrations. Although P5CS appears to be the preferential pathway for proline biosynthesis under saline conditions, there appears to be no observable pattern between different species and experimental conditions. Even when comparing salt-sensitive and tolerant cultivars, the proline biosynthetic pathways are not a useful indicator of a plant’s ability to cope with salt stress. de la Torre-González et al. [45] showed that a tolerant tomato cultivar presented a salt-induced OAT enhancement, while the proline accumulation in the sensitive cultivar was only related to reduced proline degradation. Indeed, Xue et al. [55] stated that the activation of the OAT pathway depends on the severity of the stress. Yusuf et al. [52] and Arabbeigi et al. [47] showed that sensitive and tolerant genotypes equally presented an increased activity of the proline biosynthetic pathways under salt stress (especially P5CS). However, Goharrizi et al. [46] reported that salt-tolerant wheat cultivars had higher P5CS activity induction in response to salinity than the sensitive ones. 

In this sense, proline *de novo* generation is probably not induced merely by salt exposure and cannot be taken as a sign of either tolerance or susceptibility. Moreover, proline biosynthesis can also be inhibited through negative feedback related to endogenous proline levels. Indeed, ProDH activity can decrease when plants are exposed to saline conditions, limiting the proline degradation rate and contributing to its accumulation with proline possibly binding to the γ-GK domain of P5CS and inhibiting its activity [19,56]. This might explain the unaltered P5CS activity levels described by de la Torre-González et al. [45] and Naliwajski and Skłodowska [56]. Nonetheless, the most common response when plants are subjected to salt stress is the additive effect of increased P5CS or OAT and decreased ProDH activity [48,49,50,53], indicating a tight regulation between both enzymes to ensure an adequate stress response. This was reinforced by the work of Wang et al. [51], which reported that short-term proline accumulation occurs due to the inhibition of ProDH and the activation of OAT, while in long-term salt stress or during the recovery process, biosynthetic pathways are activated to ensure a proline supply but the activity of ProDH increases or reverts to basal levels, possibly to provide energy and electrons for the respiratory chain.

#### 4.1.2. Proline Functions in Salt-Stressed Plants—Increased Tolerance or Higher Susceptibility?

Throughout the years, it has been widely believed that this overaccumulation of proline in response to stress must be related to an increased tolerance. As an osmolyte, proline’s most well-regarded function in plants under salt stress is its ability to ensure proper osmotic pressure [12,57] by limiting the accumulation of Na^+^ and Cl^-^ in the cytosol while also being able to protect plant cells from oxidative damage. However, many authors have debated if proline accumulation is a trait for an increased salt tolerance or just a symptom of stress. For instance, various authors have reported very little contribution from proline in the osmotic adjustment of salt-stressed plants (see Mansour and Ali [57] and references therein). As a more direct approach to settle this debate, there are several research papers focused on understanding the specific differences in proline metabolism between tolerant and sensitive genotypes. For instance, Forlani et al. [58] studied the response of 17 rice genotypes and found no correlation between basal or stress-induced proline levels and the tolerance index. However, a significant interaction was found between the relative increase in proline content and the ability of rice plants to tolerate salt stress. Here, the authors postulated that the role of proline in salt tolerance is not mainly related to its ability to act as an osmolyte or ROS scavenger but to activating several signaling pathways related to redox balance, NADP^+^/NADPH ratio, and the maintenance of the oxidative pentose phosphate pathway through the recently proposed proline–P5C cycle, as reviewed by Kavi Kishor and Sreenivasulu [27]. Moreover, Gharsallah et al. [59] showed that proline accumulation was higher in salt-tolerant tomato cultivars and was positively correlated with a lower Na^+^ content and improved antioxidant efficiency. Despite this, through an analysis of the existing literature it can be seen this assessment does not apply to all case studies. Even within the same species, inconsistent behavior has been observed between genotypes. While Ashrafi et al. [60] and Badran et al. [61] found that tolerant *Medicago sativa* genotypes accumulated more proline than the sensitive cultivars, X. S. Wang and Han [62] reported the opposite, pointing to proline accumulation as a symptom of injury and not a sign of tolerance. Other inconsistencies in the role of proline were found by Carpici et al. [63], Ghoulam et al. [64], Wu et al. [65], Poustini et al. [66], Kanawapee et al. [67], Lutts et al. [68], Ashraf et al. [69] and Ali et al. [70]. It becomes evident that the plant tolerance index is not always correlated to higher proline levels, but there is a clear involvement of the proline metabolism in several key pathways of the stress response. Thus, considering the important functions of this amino acid, the artificial modulation of proline levels can modify, in either a positive or negative way, the ability of plants to withstand salt stress, helping to elucidate the protective role of this compound. In addition, proline levels usually increase when salt stress mitigation strategies are employed (Appendix A), although it is still unclear whether this proline accumulation is part of the tolerance-inducing mechanism put into action by the mitigator or simply a consequence of an altered redox balance and hormonal metabolism. It is worth mentioning, however, that this is not observed in mitigation strategies for other types of abiotic stress, as can be seen later in this work (Section 5).

#### 4.1.3. Exogenous Application of Proline to Mitigate Salt Stress

As previously mentioned, proline has also been explored as a stress mitigator itself. The supplementation of proline, either by foliar spray or through irrigation, is the easiest method of inducing proline accumulation in plants, and several authors have highlighted its osmoprotectant role and benefits in improving plant tolerance to salt stress. For instance, a decrease in the Na^+^ (and Cl^−^)/K^+^ ratio was observed in several species when proline was applied exogenously [71,72,73,74,75,76,77,78,79]. These authors suggested that proline helps in avoiding Na^+^ xylem loading as well as efficiently induces the compartmentalization of Na^+^ excess, allowing for a greater influx of K^+^ and restoring the K^+^/Na^+^ ratio. Messedi et al. [80] showed that the exogenous foliar application of proline induced no differences in the Na^+^ content but enhanced the K^+^ content in *Cakile maritima* L. plants under saline stress, suggesting that proline may stimulate the root uptake of K^+^ to the detriment of Na^+^ by protecting the activity of K^+^ transporters [80]. On the other hand, Huang et al. [81] reported a decrease in Na^+^ content and in the Na^+^/K^+^ ratio in addition to an improvement in the plant’s water content, while no differences were observed in the K^+^ content. In this case, the positive role of proline seems to rely mainly on the improvement of water uptake, limiting the salt’s phytotoxic effects. This mechanism was also described by [73,79,80,82,83,84,85,86,87]. This facet of exogenously applied proline, which improves plant water relations under salinity stress, may be linked to the accumulation of some organic and inorganic compounds, such as endogenous proline, sugars, glycine betaine, and K^+^ (see Appendix A and the references therein), given that these substances can promote the adjustment of osmotic potential. Additionally, exogenous proline application further improves plant tolerance to saline stress by increasing the antioxidant potential, favoring redox homeostasis (Figure 2; Appendix A). For instance, Reza et al. [88] showed that a 15 mM proline supply to salt-stressed barley (*Hordeum vulgare* L.) plants prevented lipid peroxidation (LP) and increased catalase (CAT; EC 1.11.1.6), glutathione reductase (GR; EC 1.8.1.7), and peroxidase (POD; EC 1.11.1.7) activity. Similarly, M. A. Hossain and Fujita [89] also reported a decrease in malondialdehyde (MDA) and hydrogen peroxide (H_2_O_2_) content and an increase in glutathione (GSH) content and GR activity. Most studies also reported an increase in the activities of the antioxidant enzymes superoxide dismutase (SOD; EC 1.15.1.1) and ascorbate peroxidase (APX; EC 1.11.1.11) in addition to CAT, GR, and POD in salt-stressed plants treated with proline [71,84,90,91]; this increase is usually followed by an enhanced capacity to scavenge O_2_^•−^ and H_2_O_2_ and enhanced contents of non-enzymatic antioxidants (see Appendix A and the references therein). 

As can be seen in Figure 2, proline functions can be found at different levels of plant responses. In addition to its biochemical effects at the cellular level, proline is also recognized for affecting broader physiological mechanisms in plants such as photosynthesis, mineral nutrition, and growth. Indeed, apart from its antioxidant action, proline’s role also extends to the positive regulation of photosynthetic parameters as proline supplementation to salt-stressed plants results in higher levels of chlorophylls and carotenoids. Proline supplementation also leads to improved photosynthetic rates, such as stomatal conductance, transpiration rate, PSII yield, non-photochemical quenching, electron transport rate, and intercellular CO_2_ levels [80,83,85] (Appendix A). The positive influence of exogenous proline on the photosynthetic performance of plants seems to rely significantly on the role of proline in maintaining chloroplastic redox homeostasis and protecting the PSII from salt-induced osmotic and oxidative stress, allowing for proper electron transport activity [80]. Additionally, the positive effects of proline supply are also associated with a better nutrient uptake. In fact, proline application to salt-stressed plants increased the content of N, P, K, NO_3_^−^, NO_2_^−^, and Ca^2+^ in *Phaseolus vulgaris* [71], *Cucumis melo* [76], *Zea mays* [92], and *Sorghum bicolor* [73]. In addition to nutrient uptake, the assimilation of N can be improved by exogenous proline supply, as shown by el Sabagh et al. [93].

Together, these findings suggest that the application of exogenous proline positively affects the performance of plants under salt stress by restoring the K^+^/Na^+^ ratio, regulating water relations, maintaining the redox homeostasis, enhancing photosynthetic efficiency, and promoting a proper nutrient balance while endogenous proline levels increase. As such, it is clear that the exogenous proline supplementation helps to counteract the effects of salt stress in several plants, improving their growth and performance under these adverse conditions (Figure 2).

#### 4.1.4. Modulation of Endogenous Proline Levels in Response to Salt Stress—Genetic Engineering Approaches

Aside from understanding how plants respond to the external supplementation of proline, it is also important to unravel how genetic engineering of the proline-related pathways can be used as an efficient method of inducing plant tolerance to salt stress. For example, Guerzoni et al. [94] and Guan et al. [95] reported that transgenic sugarcane and switchgrass plants that overexpressed a form of *P5CS* accumulated more proline under salt stress and presented an increased K^+^/Na^+^ ratio in the leaf and root tissues, which was correlated with improved stress tolerance. The overexpression of *P5CS* under salt stress has also been reported to improve the maximum quantum efficiency of photosystem II, photosynthetic rate, stomatal conductance, transpiration rate, water use efficiency, and chlorophyll fluorescence in sugarcane [94] and sorghum [96]. Such protective effects of proline might be related to its ability to scavenge ROS and thus protect the thylakoid membranes against photodamage while also avoiding ROS-mediated chlorophyll degradation. Indeed, the increased growth performance in plants that over-accumulate proline is also associated with an enhanced antioxidant activity and decreased levels of ROS and MDA, highlighting the role of proline as a redox buffer and membrane stabilizer [97]. Accordingly, the knockdown of *P5CS1* and *P5CS2* leads to a reduced proline accumulation, increased ROS and MDA content, the inhibition of antioxidant enzymes, and an increased Na^+^/K^+^ ratio, affecting plant growth performance [95]. 

Reinforcing the idea that the two different isoforms of P5CS have sub-functionalization, a study by Funck et al. [23] showed that salt-stressed *p5cs1-4* mutants presented heavily impaired proline accumulation in response to NaCl, indicating that *P5CS1* contributed more to the stress-induced proline production. On the other hand, the germination and root growth of the *p5cs2-1* mutants were impaired under control condition; this was reverted by exogenous proline application, corroborating the role of *P5CS2*-induced proline production in normal plant development. The *p5cs2-1* mutants, which could still produce considerable levels of proline under stress, displayed a high salt tolerance, no signs of Na^+^ accumulation, and a much more efficient photosynthetic machinery. 

Moreover, proline-biosynthesis-related genes have binding sites for several transcription factors for which engineering has been explored to enhance salt stress tolerance, as reviewed by Zarattini and Forlani [98]. For instance, Zhang et al. [99] achieved a higher salt tolerance in soybeans when overexpressing *GmMYB84*, presenting an increased proline content, improved antioxidant efficiency, and reduced Na^+^ accumulation. Similar results were found by overexpressing *VvWRKY30* and *WRKY8* in salt-stressed Arabidopsis [100] and tomato plants [101]. J. Liang et al. [102] and Alshareef et al. [103] tested the overexpression of *JUB1* NAC transcription factors in strawberry and tomato plants, respectively. Here, J. Liang et al. [102] reported that the overexpression of the strawberry *FaNAC2* gene in *Nicotiana benthamiana* plants altered their proline metabolism, with an increased *P5CS1* and decreased *ProDH2* and *P5CDH* expression and an increased growth and antioxidant capacity under salt stress conditions. Alshareef et al. [103], found that transgenic tomato plants that overexpressed *JUB1* NAC transcription factors were more tolerant than the wild type under high salt levels. This was demonstrated by an improved biomass and increased chlorophyll content, although the increase in proline content had no effect on the leaves’ osmotic potential or on the accumulation of Na^+^ and K^+^. In contrast, the inhibition of *P5CS1* via the overexpression of *MYC2* (a bHLH transcriptional activator of abscisic acid (ABA)) in Arabidopsis [104] and the inhibition of *P5CS1* and *P5CS2* through the overexpression of *BpERF11* (an ethylene-responsive factor) in *Betula platyphylla* [105], respectively, might lead to a higher sensitivity to salt stress. Curiously, Verma et al. [104] showed that the overexpression or silencing of *MYC2* caused higher or lower salt sensitivity, respectively, and were accompanied by decreased or increased proline biosynthesis after 24 h of salt stress. However, when the stress duration was extended for one month, the *myc2* mutants were more sensitive to salinity than the wild-type plants, indicating the presence of complex cascade mechanisms that regulate long-term stress response, especially when the effects on the downregulation of ABA biosynthesis and ABA- and stress-responsive genes are considered. Accordingly, Li et al. [106] reported that although salt-stressed alfalfa overexpressing *MsHB7* (an ABA-inducible homeodomain-leucine zipper I gene) presented a higher increase in proline content (presumably due to the inhibition of ABA-mediated signaling), their tolerance to salt stress was severely reduced. Nonetheless, Zhang et al. [105] showed that salt-tolerant *erf11* mutant lines of *Betula platyphylla* still presented an improved performance after 4 weeks of exposure to salt stress. They also presented a better antioxidant efficiency and water status through the decreased stomatal aperture and increased proline levels associated with a higher expression of *P5CS1/2* and a lower expression of *ProDH* and *P5CDH*, contrasting with the opposite results that were found for the sensitive lines overexpressing *ERF11*. It is worth mentioning that the expression of P5CS can be positively regulated by ABA at expression and post-translation levels, although both ABA-dependent and independent signaling mechanisms have been reported to regulate the accumulation of proline under conditions of osmotic stress [26,98].

### 4.2. Metal-Induced Phytotoxicity

Considering the unprecedented rate of population growth in recent decades and the consequent intensification of all industrial, agricultural, urban, and technological anthropogenic activities, numerous global problems have arisen. Many of these problems are related to environmental pollution. One case that deserves special attention is the contamination of agroecosystems with concerning levels of metals [107]. Metals include high-density metallic elements and metalloids which, despite their important roles in soil constitution and nutrient cycles, tend to accumulate in these environmental matrices, causing toxicity to several organisms. 

At very low concentrations, some metals are essential micronutrients for plant development and growth as they are involved in vital redox reactions. This is the case for molybdenum (Mo), copper (Cu), nickel (Ni), manganese (Mn), iron (Fe), and zinc (Zn). On the other hand, once accumulated in the soil at levels above a certain metal-specific threshold, these micronutrients can become unsafe for plant growth, putting agriculture at as much risk as the presence of other non-essential pollutants such as lead (Pb), arsenic (As), cadmium (Cd), mercury (Hg), strontium (Sr), titanium (Ti), vanadium (V), lanthanum (La), thallium (Tl), and chromium (Cr). In this sense, the response of plants to metal-induced toxicity is not only dependent on species- and environment-related traits but also on the metal concentration, properties, and bioavailability [108,109,110]. As can be seen in Appendix A, several authors reported an overall reduction in plant growth due to metal-induced toxicity, including As, Cd, Cr, Pb and Ni, which is recognized as the consequence of a combination of the following effects: (i) the induction of nutritional imbalances due to interference with nutrient uptake and assimilation; (ii) the disruption of the photosynthetic machinery; (iii) the alteration of the plants’ water relations; and (iv) the induction of oxidative stress [107,111]. Regarding photosynthetic traits, metals can damage or inhibit the biosynthesis of pigments; disrupt the electron transport chain, CO_2_ fixation, and enzyme activity in the Calvin cycle; cause the disorganization of the granum and thylakoid membranes; and induce stomatal closure. Altogether, these alterations culminate in major growth inhibitions and yield losses, threatening food production at global level.

#### 4.2.1. Proline Homeostasis in Response to Metal Stress

When plants are exposed to toxic levels of metals, several biochemical adjustments take place within the plant cells. These adjustments include the accumulation of proline, especially in the shoot tissues [112]. As previously mentioned, such proline increases are often seen as a positive response of a plant’s defense system, suggesting that the proline is being used to counteract the effects of metal toxicity due to its general properties as an osmoprotectant, antioxidant, and metal chelator (Figure 2; Appendix A). This was the case in the reports by [113,114], who observed increases in proline levels in plants under Ni- and Mo-stress, and by Chaturvedi et al. [115], who observed increases in proline levels in response to Pb- and Zn-stress. The authors noticed that the plants’ physiological improvements seemed to be related to the higher levels of proline production. Soares et al. [116] also related the high accumulation of proline in the roots of *Solanum nigrum* with the improved resilience of the plants to Ni toxicity, and the authors Siddiqui et al. [117] and Sirhindi et al. [118] noticed that Ni-stressed wheat and soybean plants treated with gibberellin, calcium, and jasmonic acid accumulated higher levels of endogenous proline, probably as an indirect result of the positive influence of these stress mitigators in enhancing the internal defenses of the plants. Further examples of proline accumulation in response to stress can be found in several reviews [19,20,29,119,120]. Indeed, most reports link this biochemical adjustment with stress resilience, indicating that plant species that accumulate more proline under stress show better physiological performance and survival (as reviewed by Heuer [29]). Nonetheless, the stress-induced accumulation of proline in plant tissues is not always perceived as a fully positive outcome. In many studies regarding plants under metal stress, proline accumulated to levels several folds higher than in control plants while the plants still suffered considerable growth inhibitions, phytotoxicity symptoms, and severe oxidative damages that did not seem to be efficiently counteracted by endogenous proline [108,121,122,123] (Appendix A). Moreover, some authors found that proline accumulation in response to metal stress should even be seen as a signal of stress sensitivity rather than a defense mechanism [20,27,33,124]. During normal plant development, the highest proline levels are usually found in the pollen grains and seeds, where this amino acid seems to be needed to protect cells from osmotic stress during desiccation-dependent events [22]. Apart from these tissues, proline levels usually increase more in the shoots than in the roots under metal stress [112]. This is the case with tomato [122] and wheat [113], for instance. Under stress conditions, high levels of proline can also be found in the phloem, indicating that proline can also be translocated from the shoots to the roots [27].

Very curious outcomes have been noticed with respect to protein accumulation in metal-tolerant plants. It has been shown that metal-tolerant plants present overall higher constitutive levels of proline than their closest sensitive species or ecotypes, demonstrating considerable differences especially in the root tissues. In such tolerant genotypes, including *S. nigrum* (a well-known metal hyperaccumulator species—Rehman et al. [125]), the stress-induced increase in proline tend to be more tenuous and usually occurs in the roots instead of shoots [126,127]. For instance, Sun et al. [127] found that under control conditions, *S. nigrum* had higher constitutive levels of proline than the sensitive species from the same genus, *Solanum melogena*. The same authors found that proline was also more accumulated in the roots than in the shoots of *S. nigrum* during exposure to Cd [127]. Similarly, Soares et al. [116] also found a greater increase in proline in the roots than in the shoot tissues of *S. nigrum* under Ni stress. In a study using the metal-accumulator species *Atriplex halimus*, an exposure to toxic Cd levels boosted proline accumulation in both organs, but proline accumulation was boosted to a much higher extent in the roots than in the shoots [126]. In agreement with this pattern, Matsunami et al. [128] found that proline accumulation was upregulated in the roots of the stress-tolerant rice cultivar IR58. Schat et al. [129] noticed that in a metal-tolerant ecotype of *Silene vulgaris* grown under normal conditions, the constitutive proline concentration was five to six times higher than in the nontolerant ecotype. Additionally, the authors could not find increases in proline in the tolerant ecotype under exposure to toxic levels of Cu or Zn, while the sensitive ecotype accumulated high levels of proline under the same conditions of metal stress. The Cd stress only induced proline levels in the tolerant genotype at concentrations at the mM level, hundreds of times higher than in the sensitive plant in which proline was built up in response to 5 µM Cd. Cd stress disturbs water relations in plants by affecting aquaporin function, altering the structure and permeability of membranes, and affecting the transpiration rate, thus reducing the water potential. The authors suggested that proline accumulation in the shoots of the sensitive species occurred not in response to the metal per se but rather as a response to the metal-induced water deficit in the leaves [129]. Overall, metal-tolerant plants show higher constitutive levels of proline than sensitive genotypes and do not seem to undergo such drastic peaks in proline production during stress events, especially in the leaves, demonstrating more notorious increases in the roots. It could be inferred that the mechanism leading to proline accumulation in these tissues could be somehow part of a defensive strategy that makes these plants more resilient to stress. In fact, the roots are the first organ to contact soil contaminants and to perceive stressful conditions of salinity and drought; therefore, it would not be a surprise if stress-tolerant species had better root-related attributes to deal with abiotic stress. Bearing in mind how free proline can be used as an energy reservoir to provide ATP during its degradation in the mitochondria, one could infer that the accumulation of proline in the root tissues could be used as a strategic method of sustaining or stimulating root growth and elongation under stress [27], which could be used as a tolerance adaptation response to help plant roots search for unpolluted soil areas.

#### 4.2.2. Exogenous Application of Proline to Mitigate Metal-Induced Stress

The application of proline as an external stress mitigator to metal-exposed plants has shown very positive results in general, either by preventing the oxidative damage induced by these contaminants and/or through the stimulation of plant defense mechanisms to overcome metal toxicity [10,120] (Appendix A). Although there are several methods for the exogenous application of this amino acid, foliar spraying and seed priming are the most effective [10,120]. The foliar application of proline to maize plants grown under Cd and Pb improved growth performance, delayed leaf and flower senescence, and improved photosynthesis, antioxidant enzyme activity, and levels of phenolic compounds, leading to an increased seed yield [119,130]. The exogenous application of proline to an olive tree and wheat plants promoted the uptake of nutrients under stress induced by Cd and Cu, respectively [131,132]. On one hand, proline contributes to the protection of cells from the toxic effects of metals; on the other hand, proline sometimes also seems to play a role in metal detoxification. Examples include studies by several authors [131,132,133,134] in which the positive influence of proline was partially due to restricting the uptake or translocation of the respective metal (Appendix A). Chandrakar et al. [135] reported that the proline-mediated reduction of metal root-to-shoot translocation seemed to be related to the induced synthesis of phytochelatins, which are well-recognized molecules for their metal-chelating properties and their role in metal detoxification. Curiously, as can be seen in Appendix A, some works reported that a supply of proline sometimes stimulated the uptake and increased the accumulation of metals in plants [136,137,138]. For instance, the work by Xu et al. [137] on *S. nigrum* under Cd toxicity showed that the application of proline enhanced the uptake of Cd by a phytochelatin-mediated mechanism. The authors suggested that proline, by helping to maintain the redox balance in the cytoplasm, alleviated the GSH depletion, favoring the synthesis of phytochelatin (see review by X. Liang et al. [31]). In an aside, exogenous proline has been shown to exert an effect on the levels of GSH, increasing the GSH/GSSG ratio in the leaves and roots of *B. juncea* [139], validating the indirect effect of proline in stimulating another multifunctional compound, GSH, and strengthening downstream mechanisms of stress tolerance. In fact, Xu et al. [137] suggested that in this way, the conjugation of Cd with phytochelatin would be more easily sequestered in the vacuoles, enhancing the Cd accumulation ability and the phytoremediation potential of these plants [137]. Yu et al. [138] also suggested that proline might have prevented Cr (VI) from reaching the shoot tissues by “storing” it in the root apoplast. Indeed, the compartmentalization of metals in root cell walls is an effective strategy for avoiding metal toxicity and is commonly implemented by metal-accumulator species, conferring the protection of the photosynthetically active tissues and decreasing metal-induced oxidative damages [140]. Regarding the proline-mediated stimulation of metal uptake, Ullah et al. [141] found that the accumulation of Pb and Cr was enhanced in plants that accumulated higher levels of endogenous proline, especially in the roots. 

In terms of photosynthesis, the application of proline has also been shown to attenuate the negative effects induced by metals [83,119,142,143]. Exogenously applied proline was reported to stabilize not only the electron transport chain between cell membranes and mitochondria but also the 3D structure of proteins, preserving the activity of important, photosynthesis-related enzymes such as ribulose-1,5-bisphosphate carboxylase-oxygenase (RuBisCO; EC 4.1.1.39) [119]. Furthermore, proline promoted stomatal opening in plants subjected to Ni and Pb by blocking the connection of ABA and specific proteins on cell guard membranes [143] and by increasing the levels of K^+^ on guard cells [130], respectively, thus favoring the stomatal opening. In eggplants (*S. melongena*) exposed to As, the exogenously applied proline seemed to act as an electron donor for PS II [143]. The CO_2_ assimilation rate in metal-stressed plants was also restored upon the addition of exogenous proline by an increase in photosynthetic gas exchange attributes and by enhanced biosynthesis and/or the decreased degradation of chlorophylls and carotenoids [139,142]. 

Regarding the redox homeostasis, the application of proline, either by foliar spraying or through irrigation, usually leads to an enhanced antioxidant potential [120,130,135,139,142,144,145,146]. This happens not only due to the direct role of proline as an antioxidant per se but also because of its ability to trigger other defensive pathways, including antioxidants. For instance, the contents of GSH, phenolic acids, ascorbic acid, flavonoids, and polyamines have been reported to increase in response to proline treatment in *Coriandrum sativum, Zea mays, Olea europaea,* and *Pisum sativum* under Pb, Hg, or Ni stress [83,120,146,147,148,149]. The gene expression and activity of antioxidant enzymes such as SOD, CAT, APX, and POX [83,130,135,136,145,148] were also reported to increase, helping to avoid the excessive build-up of ROS [97]. In fact, exogenous proline could decrease the levels of H_2_O_2_ and/or O_2_.^-^ in pea, brown mustard, common bean, eggplant, olive tree, brown mustard, pigeon pea, soybean, wheat, and black nightshade plants exposed different metals such as Ni, Cd, Pb, Se, and As [83,135,137,139,142,143,144,145,146,149,150]. In plants exposed to Cd, Ni, Pb, and Se, proline treatment led to lower degrees of oxidative damage, observed by the decline in contents of MDA, thiobarbituric-acid-reactive substances (TBARS), and electrolyte leakage (EL) [83,120,144,145,146]. Furthermore, proline has been reported to stabilize membranes, making them less porous and leaky and less susceptible to the oxidative damage caused by these contaminants [135,144,150].

### 4.3. Exposure to Xenobiotics

#### 4.3.1. Xenobiotic Exposure and Proline Metabolism—Is There a Connection?

Environmental degradation caused by the accumulation of organic compounds called xenobiotics is paradoxically an old yet emergent issue currently affecting ecosystem dynamics [151,152]. Indeed, due to long-lasting and recent anthropic activities such as those related to the industry, nanotechnology, healthcare, cosmetics, and agri–food systems, soils and surface waters are becoming sinks of innumerous synthetic compounds, including drugs and pesticides, jeopardizing the normal functioning of agro-ecosystems [153].

With the exception of herbicides, most of these products are not designed to directly affect plant cells; therefore; plants are often overlooked in risk assessments of xenobiotics when compared to animal systems. Still, especially over the last 10–15 years, considerable effort has been placed on understanding the effects of different kinds of xenobiotics on plant metabolism [154,155,156,157,158,159,160,161]. Given their generalized use and their recognition as emergent pollutants, attention has been mainly centered on pesticides and pharmaceuticals [154,155,156,160,162,163,164]. In opposition to several inorganic contaminants such as essential metals (e.g., Cu and Zn), plants do not usually present specific mechanisms to uptake organic contaminants; nonetheless, they are able to uptake them using non-selective mechanisms and can accumulate these compounds to a certain extent, leading to a series of physiological and biochemical disturbances that usually affect growth and productivity. Once accumulated inside plant tissues, xenobiotics can interact with several cellular players such as proteins and nucleic acids, inducing a cascade of downstream effects which usually involve the disruption of redox homeostasis [155,165]. For instance, Gomes et al. [166] and Soares et al. [167,168], when reporting the biochemical and cellular effects of glyphosate (GLY) exposure on willow (*Salix miyabeana* Seemen) and tomato (*Solanum lycopersicum* L.) plants, revealed that the plants underwent a state of oxidative stress, measured by the increase in LP and the overaccumulation of ROS, while also experiencing changes to the antioxidant system, including an increase in proline levels. The upregulation of proline production has also been reported in response to pharmaceuticals [158], although proline metabolism seems to fall into a species-specific regulation in response to these organic pollutants. As reported for other types of stress previously detailed in this review, proline is often considered a good biomarker of xenobiotic exposure [169,170]. Works performed with several plant models ranging from monocots to dicots revealed that proline accumulation is usually modulated by the presence of these organic compounds, although a constant response cannot always be found (Appendix A).

#### 4.3.2. Impacts of Pharmaceuticals-Induced Stress on Proline Endogenous Levels

Of the wide range of organic contaminants found in soils and surface waters, pharmaceuticals and cosmetic-related compounds are among the most common due to their widespread utilization for both human and animal health care purposes [164]. Usually, plants contact these pollutants directly via the roots, where these substances exert their first toxic symptoms. When studying the phytotoxic hazards of several drugs on alfalfa plants, individually or combined (diclofenac, sulfamethoxazole, trimethoprim, and 17a-ethinylestradiol, all tested), Christou et al. [171] revealed distinctive responses depending on the analyzed organ (shoot or root) and the chemical itself. Interestingly, although no substantial effects were observed in plant growth performance, substantial changes were recorded subcellularly, probably because of xenobiotic bioaccumulation and an interference with the normal cellular metabolism. In general, alfalfa seedlings exposed to these contaminants (especially diclofenac), individually or combined, demonstrated a lowered accumulation of proline in both the shoots and roots and an aggravation of lipid peroxidation in the roots [171]. In fact, the modulation of proline content in response to these types of contaminants has also been observed in other works [157,158,172]. Martins et al. [157] reported that the accumulation of proline in barley plants followed a dose-response manner, being noticeable only for higher tested concentrations of diclofenac. Moreover, it should be noted that, upon exposure to diclofenac, the maintenance of proline levels reported by Pawłowska et al. [172] was not followed by a rise in lipid peroxidation or H_2_O_2_. In contrast, when studying the toxic effects of diclofenac on tomato seedlings, Sousa et al. [158] observed a boost in proline levels paired with an overaccumulation of H_2_O_2_, which culminated in increased membrane damage in the roots. Given the high variability found in the literature, it appears that, at least with respect to diclofenac-induced stress, proline metabolism does not seem to be a key signature point and most probably occurs as a downstream consequence caused by, for instance, xenobiotic-induced effects on enzyme activity. Based on the assumption of Christou et al. [171], the modulation of proline levels in response to diclofenac may arise as a result of xenobiotic interference with its metabolic pathway, either by favoring proline catabolic enzymes and/or inhibiting those responsible for its biosynthesis [171]. This hypothesis was further supported by Martins et al. [157], whose work revealed that under diclofenac stress, the production of proline by tomato plants is ensured by an alternative pathway related to the GS/GOGAT cycle as diclofenac suppressed glutamine synthetase (GS; EC 6.3.1.2) but enhanced glutamate dehydrogenase (GDH; EC EC 1.4.1.2) as an additional source of glutamate, the main precursor of proline in conditions of stress. Proline accumulation as a response to paracetamol, one of the most commonly utilized drugs worldwide, has also been reported [173]. Barley plants growing in paracetamol-contaminated soils showed no clear signs of oxidative damage, despite the strong growth inhibitions observed [173], which the authors suggested were in relation to the observed accumulation of proline, especially in the leaves, which seemed to have played a key role in protecting the cell membranes. Curiously, in that paper, upon treatment with silicon dioxide nanomaterials (nano-SiO_2_), proline levels were not changed in the barley leaves but were decreased in the roots when compared to plants that were only exposed to paracetamol, suggesting that proline accumulation under single exposure was indeed a protective mechanism and was no longer required after the co-treatment [173]. An upsurge of proline does not always seem to be accompanied by an increased tolerance. A former study with two macrophytes (*Lemna minor* L. and *L. gibba*) established a correlation between the accumulation of proline and the degree of tolerance—while *L. minor*, which is particularly sensitive to acetaminophen, did not alter the proline content, *L. gibba* curiously decreased the levels of this amino acid and showed no signs of oxidative damage [174].

Although it has been explored less, the phytotoxicity of other pharmaceuticals, such as tetracycline and naproxen, has also been assessed. An enhanced accumulation of proline took place in barley leaves, especially when exposed to high concentrations of naproxen (1000 mg kg^−1^). This was paired with a clear oxidative disruption and increases in ROS and MDA content [172]. Regarding tetracycline, in a work performed with two aquatic plants (*Hydrocharis dubia* (Bl.) Backer and *Trapa bispinosa* roxb.), the authors observed a time- and species-dependent response of proline homeostasis [175]. According to the authors, the ability of *T. bispinosa* plants to maintain higher levels of proline (along with protein) during long-term exposure to tetracycline should be interpreted as an indicator of the higher stress tolerance of *T. bispinosa* in relation to *H. dubia*, which is associated with the major role of proline as an efficient osmoprotectant [175].

#### 4.3.3. Impacts of Pesticide-Induced Stress on Proline Endogenous Levels

Another class of xenobiotics that has been gaining attention from both ecotoxicological and phytotoxic researchers are the agrochemicals. Indeed, a growing number of studies have been unravelling the effects of several pesticides used in agriculture on the growth and redox homeostasis of non-target plant species through the analysis of ROS production and antioxidant metabolism [155,176,177,178,179]. Such studies have been conducted for the insecticides deltamethrin, imidacloprid, phosalone, emamectin benzoate, α-cypermethrin, chlorpyrifos, omethoate, and dimethoate; the fungicides metalaxyl, acetophenone, chlorothalonil, tricyclazole, and thiabendazole; and herbicides such as bentazon, prometryn, butachlor, paraquat, fluroxypyr, clopyralid, atrazine, and glyphosate. A thorough comparison of the outcomes of these articles highlights, in most cases, the accumulation of proline as a common effect of pesticide-induced stress, regardless of the xenobiotic and plant species considered (Figure 2). At the same time, these increased proline levels are often accompanied by enhanced activities of GST and SOD enzymes, despite the harsh oxidative damages and growth inhibitions observed in plants under pesticide-induced stress. Altogether, these results suggest that the accumulation of proline in plants exposed to pesticides seems to be tightly associated not with its antioxidant properties of membrane stabilization and ROS scavenging but with its action as a signaling molecule, as noted by Shopova et al. [180]. From this perspective, the increased levels of proline seem to act as a stress signal that triggers the expression of genes related to the antioxidant defense system and the xenobiotic detoxification pathway, counteracting two of the most common consequences of pesticide exposure: the overproduction of ROS and the bioaccumulation of pesticides in plant tissues, respectively. Although the existence of a direct interaction between proline accumulation and pesticide toxicity is hard to establish, it has been shown that the overexpression of a bacterial D-amino acid oxidase (*DAAO*), a glyphosate-degrading enzyme, in *Arabidopsis thaliana* seedlings increased plant resistance to the herbicide, lowering the overaccumulation of ROS and significantly reducing the levels of proline in comparison to wild-type plants [181], thereby allowing for plant survival upon exposure to glyphosate.

#### 4.3.4. Xenobiotic Exposure vs. Mitigation Strategies—Is There Room for Proline Action?

Given the important role that different pharmaceuticals and pesticide classes play in human health and agri–food production, respectively, more than characterizing their negative effects on plant physiology, it is crucial to develop strategies to alleviate the stress induced by these xenobiotics in order to minimize their impact on crop yield. In comparison to other types of abiotic stresses, there are few strategies targeting the alleviation of xenobiotic-induced stress in plants, especially targeting specific cellular and biochemical mechanisms involved. Even so, several authors have tested multiple alternatives that could be implemented in real agronomic scenarios by either seed or foliar application to help plants deal with the environmental presence of different organic pollutants. Although the outcomes are not always straightforward, the beneficial effects observed are usually associated with an overall improvement of the antioxidant system and detoxification responses as oxidative stress is often the common language between plant cells and stress exposure.

In general, the exogenous treatment of plants exposed to different herbicides with salicylic acid, nano-SiO_2_, Na_2_SiO_3_·5H_2_O, sodium nitroprusside (SNP), or proline led to the reduction of endogenous proline levels in comparison with the stress factor alone, which demonstrated a significant increase in proline content in relation to the control (Figure 2; Appendix A). However, the mechanisms behind this common effect seem to rely mainly on the type of treatment applied rather than the plant species or the class of xenobiotics. For instance, while it has been reported that the root uptake of herbicides such as glyphosate can be reduced upon treatment with salicylic acid, the application of Si-containing compounds, NO donors (SNP), and salicylic acid once more efficiently alleviate herbicide-induced stress, likely due to the stimulation of the antioxidant system through both increased levels of carotenoids and GSH and the activation of the enzymes SOD, CAT, APX, and GST. Moreover, upon exogenous administration of different phytoprotective compounds, such as Si-containing formulations, salicylic acid, and NO-donors, the glyphosate-induced accumulation of proline is frankly repressed, which is accompanied by a better growth performance and an increased activity of antioxidant and detoxification mechanisms [167,168,182]. Altogether, these observations strengthen the premise that proline is used as an early warning signal of herbicide exposure, mediating a series of stress responses rather than tolerance mechanisms. 

When exploring the exogenous application of proline (Appendix A), Ould Said et al. [179] suggested that the external supply of proline inhibited the accumulation of endogenous proline in plants exposed to the herbicide, promoting the activation of a proline catabolic pathway to degrade its excess. This was evidenced by an increase in the activity of ProDH. Despite the concomitant and generalized reduction of the activity of other components of the antioxidant system, plants exposed to bentazon showed a better growth performance, highlighting the powerful antioxidant activity of proline. In contrast, the foliar application of brassinosteroids to mitigate pesticide-induced stress did not cause a consistent change in proline levels compared to the stress factor alone [180,183]. In fact, both studies concluded that brassinosteroid treatment activated the pesticide detoxification pathway, with GST playing a leading role in this outcome.

## 5. Concluding Remarks and Future Perspectives

Extensive reviews on the roles of proline in plants under abiotic stress emphasize its positive effects as a key player in inducing plant stress tolerance [10,12,13,14,15,16,31], recognizing its clear involvement in osmoprotection, the scavenging of ROS, redox buffering, enzyme protection, membrane stabilization, the activation of antioxidant mechanisms, and the stimulation of metal chelation. In general, the exogenous application of proline as a stress mitigator has been revealed to be a promising strategy for enhancing the resilience of plants to stress factors derived from soil degradation effects, including salinity and contamination by metals and xenobiotics (Figure 2; Appendix A). However, there were also a considerable number of studies that could not find such optimal results using proline as a stress mitigator [19,20], and other studies even reported toxic effects of exogenous proline [184]. Moreover, despite a significant amount of evidence regarding the role of proline accumulation in mediating stress tolerance, there is still much discussion on whether exacerbated levels of proline can be directly correlated with improved performance. Still, in most cases, proline levels are significantly modulated by stress, meaning that this amino acid must be playing a role in the stress response. Some authors suggest that the occurrence of peaks in proline accumulation during stress, especially in the shoot tissues, occur as a symptom of high sensitivity to stress, serving as an alarm that signals the imposition of stress without necessarily being involved in cell protection [124].

Future studies should be careful to discern the patterns of proline accumulation in stress-sensitive vs. tolerant genotypes. They should also discern the roles and mechanisms affected by proline when this amino acid is accumulated in sudden peaks upon the imposition of stress from when this stress-induced accumulation occurs more gradually over time. The quantification of proline should not be used as a mere indicator of stress sensitivity or stress defense, and proline quantification should always be accompanied by assessments of other biochemical and physiological endpoints. Moreover, the relative accumulation of proline in shoot and root tissues and the occurrence of proline translocation events between plant organs should be taken into consideration. Additionally, assumptions about the roles of endogenous proline should not be made based on the action of proline that is artificially accumulated through genetic engineering or exogenous application. Indeed, the importance of proline in plants under abiotic stress is still very much a matter of debate. In this sense, proline metabolism and homeostasis in response to stress should be more carefully studied, including further conditions of abiotic stress and times of exposure, and taking specificities of stress, species, tissue, and organs into account. An investigation of the effects of abiotic stress on the kinetics of proline metabolism with respect to the enzymes related to proline synthesis and catabolism is highly encouraged.

## Figures and Tables

**Figure 1 antioxidants-12-00666-f001:**
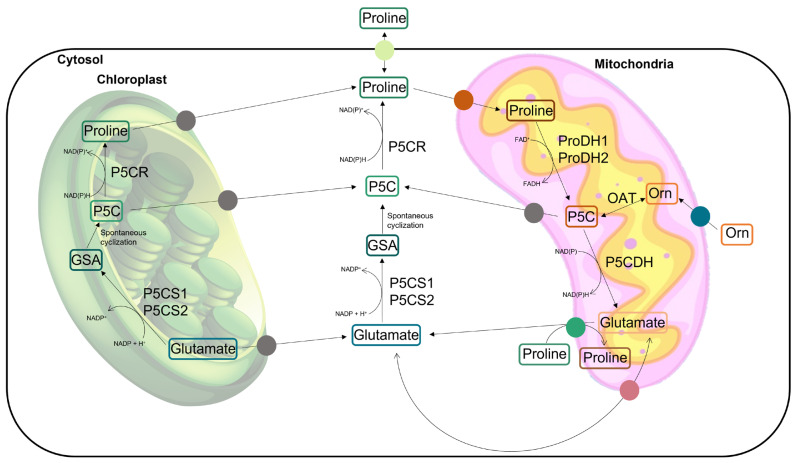
Overview of the main metabolic pathways responsible for proline biosynthesis in the cytoplasm and/or chloroplast and for proline catabolism in the mitochondria. The protein transporters involved are highlighted in colored circles. Putative transporters are presented as grey circles. P5CS: Δ1-pyrroline-5-carboxylate-synthetase; P5CR: Δ1-pyrroline-5-carboxylate-reductase; GSA: glutamic-5-semialdehyde; P5C: pyrroline-5-carboxylate; ProDH: proline dehydrogenase; P5CDH: pyrroline-5-carboxylate dehydrogenase; and OAT: ornithine δ-aminotransferase.

**Figure 2 antioxidants-12-00666-f002:**
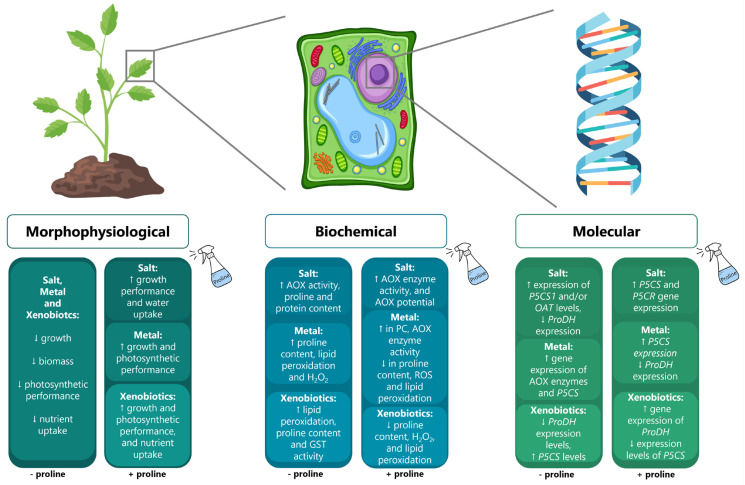
Common effects observed in plants exposed to conditions of salinity, metal contamination, and pollution by xenobiotics at morphophysiological, biochemical, and molecular levels (left boxes) in comparison to the response of stressed plants to the exogenous application of proline (right boxes). ↑ and ↓ symbolize the reported increase or decrease of a certain parameter, respectively.

## Data Availability

The data presented in this study are available in the article and Appendix A.

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
