# Peer review of "Accumulation of Proline in Plants under Contaminated Soils—Are We on the Same Page?"

_antioxidants, 2023, doi:10.3390/antiox12030666_

Round 1
Reviewer 1 Report
The paper covers the topic of the accumulation of proline in plants under stress by an intriguing point of view about its role, that could be a way to improve plant tolerance or simply an indicator of stress. It is well focused and structured and allows the reader access the pertinent information. The scientific quality is very good and the topic is of current interest under a scenario of increasing plant abiotic stresses. It can be considered self-supporting, but some minor revisions could furtherly improve it: in particular, other important compounds playing the same roles in plants should be compared or at least mentioned, such as betaine and starch. For example, it’s hard to assume that plant synthesizes and accumulates proline in roots as energy and carbon source to sustain and stimulate roots growth and elongation, as usually starch does (see lines 500-505).
Minor revisions:
Figure 1 readability should be improved. Figure caption should be more explanatory and completed with full explanation of abbreviations in metabolic pathways.
Line 75: what does “incorporated” mean?
Line 137: the term “potential” associate to “water uptake” can be misleading. Please rephrase.
Line 201: eliminate “O”.
Lines 331-332: Explain better how “assimilation of N” can be “in addition” to nutrient uptake.
Line 492: Explain better why water deficit in leaves can be a consequence of metal toxicity.
Lines 593-594: The sentence “plants do not usually present specific mechanisms to uptake organic contaminants” is a little bit astonishing: xenobiotic uptake in plants has been largely studied, and also recently reviewed (J ones, B.M., Collins, C.D. (2020). Measuring and Modelling the Plant Uptake and Accumulation of Synthetic Organic Chemicals: With a Focus on Pesticides and Root Uptake. In: Ortega-Calvo, J.J., Parsons, J.R. (eds) Bioavailability of Organic Chemicals in Soil and Sediment. The Handbook of Environmental Chemistry, vol 100. Springer, Cham. https://doi.org/10.1007/698_2020_591). Maybe the authors could rephrase the sentence, or explain better their thought (or eliminate the sentence).
Line 644: correct uppercase after full stop (Proline).
Lines 705-707: the sentence is questionable, too generic and not supported.
Author Response
We, the authors, are grateful to the reviewer for the rapid and detailed evaluation of our manuscript. We have addressed their valuable comments and corrections, and alterations have been included in the text of the revised manuscript, using the tracked changes tool.
The following minor corrections were done: The caption of Figure 1 has been improved according to the reviewer’s suggestions. The word “incorporated”, in line 75, was replaced with “biosynthesized”. The term “potential”, in line 137, was removed to avoid confusion. “O” was eliminated from line 201, as it was a gaffe. The correct uppercase for Proline was used in line 644.
In response to the reviewer's comment: "Lines 331-332: Explain better how “assimilation of N” can be “in addition” to nutrient uptake.", we would like to clarify that N uptake and N assimilation were not used as synonyms, given that these are widely accepted as distinct processes in plant tissues. While the uptake refers to the entry of N through the roots, N assimilation requires the reduction of nitrate to ammonium, followed by ammonium assimilation into amino acids, mainly through the GS/GOGAT cycle.
In response to the reviewer’s comment “Line 492: Explain better why water deficit in leaves can be a consequence of metal toxicity.”, we included information on the effects of metals on plant water relations. In fact, Cd stress, similarly to other metal stresses, commonly disturbs the water relations, inducing water deficit in plant cells. For instance, metal stress can lead to impairment of aquaporin function and to the disruption of the plasma membrane structure with a consequent decrease of the water permeability of the cells. Besides, metal stress affects the transpiration rate, reducing water potential.
Regarding the reviewer’s comment “ Lines 593-594: The sentence “plants do not usually present specific mechanisms to uptake organic contaminants” is a little bit astonishing: xenobiotic uptake in plants has been largely studied, and also recently reviewed (J ones, B.M., Collins, C.D. (2020). Measuring and Modelling the Plant Uptake and Accumulation of Synthetic Organic Chemicals: With a Focus on Pesticides and Root Uptake. In: Ortega-Calvo, J.J., Parsons, J.R. (eds) Bioavailability of Organic Chemicals in Soil and Sediment. The Handbook of Environmental Chemistry, vol 100. Springer, Cham. https://doi.org/10.1007/698_2020_591). Maybe the authors could rephrase the sentence, or explain better their thought (or eliminate the sentence).”, what we aimed to state in this sentence was that although the uptake of organic compounds occurs, this is mainly done via nonselective mechanisms. Many organic nitrogen transporters have low selectivity and therefore are capable of uptaking organic compounds due to the similarity of their nitrogen-containing functional groups. Also it is known that passive diffusion, use of non-specific protein transporters and other ways of entry are used by organic contaminants, depending on their charge, lipophilicity, molecular mass, number of H-bond donors, number of rotatable bonds, and similarity with functional groups, etc. A slight modification was made to the text to clarify this.
In response to the reviewer's comment: “Lines 705-707: the sentence is questionable, too generic and not supported.”, which regards the sentence “In comparison to other types of abiotic stresses, strategies targeting the alleviation of xenobiotic-induced stress in plants, especially targeting specific cellular and biochemical mechanisms involved, are only a few.”, we would like to state that this sentence is part of the introductory paragraph to the present section 4.3.4, which is why it is so "generic". Although it is a fact that the number os studies regarding stress imposed by exposure to xenobiotics is far lower than that for other types of abiotic stress, many of the existing reports on this matter are indeed discussed in the present work. Thus, this sentence is supported by the following paragraphs in which such studies are described in more detail.

Reviewer 2 Report
The manuscript from Spormann et al deals with the classical question on whether stress markers, such as proline, are actually making plants more resistant or they are just a consequence of sensitivity. This is an interesting topic and always on debate, as there are results confirming one hypothesis and the opposite. Authors make an extensive and clear review with recent results on the field, the text is well written and figures clear. Therefore, I recommend its publication in this journal in the present form.
Author Response
We, the authors, are grateful to the reviewer for the rapid evaluation of our manuscript. We appreciate the reviewers’ compliments on our manuscript and are pleased to know that it met the reviewer’s standards for recommendation for publication.
Reviewer 3 Report
This paper is a review that describes the proline functions in plants under stress conditions. The content is well organized. I have three minor points.
1) I am happy if the authors describe that the proline biosynthetic pathways are ABA-dependent or independent. As you know, ABA is one of the important molecules under stress conditions.
2) In Fig. 2, the authors set the two boxes containing words that explain the common effects of proline exposure. I am glad that the sizes of the words are bigger. Moreover, please add the explanation that the left box is through irrigation and the right box is by spray application (is this correct?).
3) P5, line 201, "an 0 exacerbated" is "an exacerbated"?
Author Response
We, the authors, are grateful to the reviewer for the rapid and detailed evaluation of our manuscript. We have addressed their valuable comments and corrections, and alterations have been included in the text of the revised manuscript, using the tracked changes tool.
In response to the reviewer’s comment “I am happy if the authors describe that the proline biosynthetic pathways are ABA-dependent or independent. As you know, ABA is one of the important molecules under stress conditions.”, we added the following information: “It is worth mentioning that the expression of P5CS can be positively regulated by ABA at expression and post-translation levels, although both ABA-dependent and independent signaling mechanisms have been reported to regulate the accumulation of proline under conditions of osmotic stress [26,98].” at the end of section 4.1.4.
Regarding the second comment “In Fig. 2, the authors set the two boxes containing words that explain the common effects of proline exposure. I am glad that the sizes of the words are bigger. Moreover, please add the explanation that the left box is through irrigation and the right box is by spray application (is this correct?).”, we would like to clarify that for each “level” (morphophysiological /biochemical/ molecular) the box on the left describes the effects observed in plants in response to stress by salinity, metals and xenobiotics, while the box on the right shows the effects observed in response to exogenous proline application in plants under these same stress factors. We have made changes to Figure 2 and respective caption, increasing the font size and clarifying the difference between the left and right boxes to avoid this confusion.
In line 201, “O” was eliminated, as it was a gaffe.
